# Tumor-skin invasion is a reliable risk factor for poor prognosis in superficial soft tissue sarcomas

Tadashi Iwai◉*, Manabu Hoshi, Naoto Oebisu, Naoki Takada, Yoshitaka Ban, Hiroaki Nakamura

Department of Orthopedic Surgery, Osaka Metropolitan University Graduate School of Medicine, Abeno-Ku, Osaka, Japan

* qq329xpd@opal.ocn.ne.jp

## Abstract

### Introduction

Superficial soft tissue sarcomas are often left untreated unless they invade the skin and skin ulcers manifest. Progressive sarcomas frequently result in dismal oncological outcomes despite multidisciplinary treatment. This study aimed to identify prognostic factors for superficial soft tissue sarcomas.

### Materials and methods

This study retrospectively analyzed the clinicopathological data of 82 patients with superficial soft tissue sarcomas treated between August 2003 and December 2020 at our institution. A superficial soft tissue sarcoma was defined if the percentage of the area occupied by the tumor in the assessed region (skin, subcutaneous) was more than 50%. Age, sex, location, tumor size, tumor-skin invasion, tumor grade, and distant metastasis at initial diagnosis were evaluated as potential prognostic factors. Cox proportional hazards regression models were used to identify the prognostic factors. Five-year survival rates were assessed by the Kaplan-Meier method.

### Results

The mean follow-up time was 60.1 months. The 5-year overall survival, 5-year local recurrence-free survival, and 5-year metastasis survival rates were 76.4%, 60.6%, and 71.0%, respectively. Univariate analysis showed significant relationships between poor prognosis and tumor size ≥5 cm, distant metastasis at initial diagnosis, and tumor-skin invasion. In the multivariate analysis, only the tumor-skin invasion was associated with worse overall survival.

### Conclusions

Superficial soft tissue sarcomas have biologically been considered a separate category due to their better prognosis. In this study, the tumor-skin invasion was the only significant factor

**Data Availability Statement:** All relevant data are within the paper and its Supporting Information files.

**Funding:** The authors received no specific funding for this work.

**Competing interests:** The authors have declared that no competing interests exist.

associated with a poor prognosis. Therefore, all superficial soft tissue sarcomas without tumor-skin invasion should be treated as early as possible.

## Introduction

Soft tissue sarcomas occur infrequently, accounting for only 1% of all malignant tumors [1], and superficial soft tissue sarcomas are less common as compared with deep sarcomas [2]. Tumors that continue to grow in size (to >5 cm) are painful, deeply situated, and may be malignant [3]. Superficial soft tissue sarcomas are generally smaller than deep sarcomas and are associated with lower rates of distant metastasis and higher rates of disease-free survival [4]. However, when the tumors occur superficially, that might distort the clinical decision-making of the oncologists and thus, they might not suspect sarcomas, resulting in a misdiagnosis [5].

Recently, some superficial soft tissue sarcomas have been left untreated in the absence of skin invasion and skin ulcers. Ulceration caused by skin invasion of malignant tumors is known as a 'malignant wound' [6, 7]. Previous reports have indicated that the presence of malignant wounds is associated with poor prognoses in carcinomas [8]. Okajima et al. reported that malignant wounds in soft tissue sarcomas were statistically significantly associated with poor prognoses [9].

There are currently no standard criteria for treating and evaluating superficially located tumors even though cutting-edge research indicates that an assessment of the major histocompatibility complex is crucial for the clinical outcome of sarcoma immunotherapy [10]. In this study, we evaluated the location of lesions by evaluating each magnetic resonance (MR) image separately and defined a tumor as superficial soft tissue sarcoma if the percentage of the area occupied by the tumor in the assessed region (skin, subcutaneous) was more than 50% (Fig 1A).

Moreover, few reports have focused on superficial soft tissue sarcomas with the goal of analyzing their associations with tumor-skin invasion outcomes, including with respect to malignant wounds and poor prognoses. Therefore, this study aimed to identify the prognostic factors for superficial soft tissue sarcomas and to analyze the correlations between tumor-skin invasion and poor prognoses.

## Methods

### Data aggregation

Clinical characteristics were retrospectively collected from 82 patients (48 men and 34 women, mean age at initial consultation: 63.6 years [range 21–87 years]) treated for superficial soft tissue sarcomas at the Department of Orthopaedic Surgery (Osaka Metropolitan University Hospital) between August 2003 and December 2020. The present study was approved by the Institutional Review Board of Osaka Metropolitan University Graduate School of Medicine and was performed in accordance with the ethical standards laid down in the Declaration of Helsinki (no. 4394). The subjects included in the study provided informed consent prior to their inclusion in the study.

### Patient diagnoses

Radiological evaluations using X-rays, local computed tomography (CT), and MRI with or without gadolinium enhancement were conducted in all patients. Both fluorodeoxyglucose

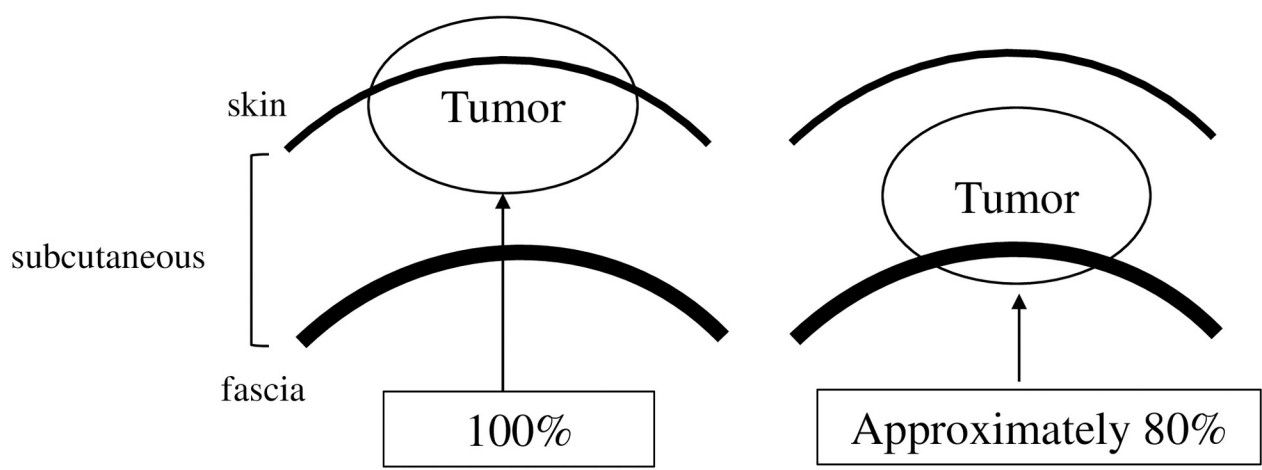

**a**

# Superficial soft tissue tumor

The area occupied by the tumor in the skin and/or subcutaneously: >50%

**b**

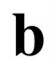

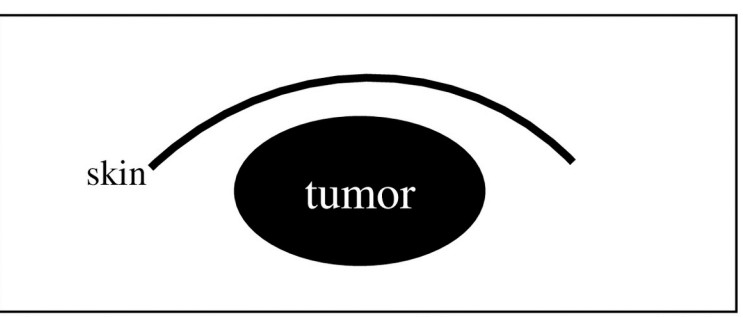

No skin invasion

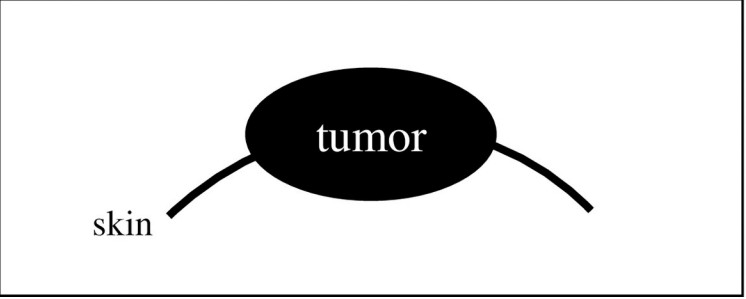

Skin invasion

**Fig 1. Magnetic resonance imaging (MRI) analysis.** (a) the definition of superficial soft tissue tumors, and (b) the relationship between tumor characteristics and skin parameters (no skin invasion vs. skin invasion).

positron emission tomography combined with CT and chest CT were performed to assess distant metastases. If a malignant tumor was suspected radiologically, the patient was recommended to undergo a biopsy. Based on standard diagnostic criteria for soft tissue sarcoma subtyping, all specimens were diagnosed by two pathologists specializing in sarcoma pathology [11]. Histological grade was evaluated in accordance with the Federation Nationale des Centres de Lutte Contres le Cancer grading system for soft tissue sarcomas [12]. Grade 1 sarcomas were classified as low-grade, and grade 2–3 sarcomas were classified as high-grade. Clinical staging was determined according to the American Joint Committee on Cancer (8th edition) guidelines with respect to soft tissue tumors and bone cancer [13].

## MRI analysis

MRI was performed in all patients prior to needle biopsy and/or excision biopsy. MRI information was evaluated by two orthopedic oncologists. Superficial soft tissue sarcomas were carefully defined based on their location (skin and/or subcutaneous). After evaluating the location of the lesions in each MRI image, a tumor was defined as a superficial soft tissue sarcoma if the percentage of the area occupied by the tumor in the assessed region (skin, subcutaneous) was >50% (Fig 1A). Tumor size (length x width x height) was measured exactly and the maximum diameter of the tumor was recorded. Lesions were categorized into two groups: tumors measuring <5 cm and tumors measuring ≥5 cm. To further classify relationships with sarcoma outcomes, tumors were also divided into two groups depending on if the tumor had invaded the skin or not (Figs 1B and 2).

## Parameters

The following pre-treatment parameters were evaluated: age, sex, tumor location, tumor size, histological diagnosis, histological grade, distant metastases upon initial diagnosis, tumor-skin invasion, and oncological outcomes. Tumor size, location, distant metastases upon initial diagnosis, and tumor-skin invasion parameters were estimated using MRI and/or CT. We assessed the surgical margins of the specimens based on the guidelines specified by the Japanese Orthopaedic Association [14].

## Patient follow-up

After treatment, patients were regularly followed up at 3-month intervals. Local examination, chest radiography and/or CT were performed for the first 2 years post-treatment. From 3 to 5 years after treatment, patients were followed up every 6 months and/or annually according to physician judgement. MRI examinations were conducted to detect post-operative local recurrence every 6 months for the first 3 years. Follow-up time was defined as the interval from the first surgery to the last follow-up.

## Statistical analysis

Post-surgery survival curves for patients with superficial soft tissue sarcoma were plotted using the Kaplan-Meier method [15]. Log-rank tests were used to compare survival times between the two groups [16]. Univariate analysis was conducted using the log-rank test. Multivariate analysis was implemented using the Cox proportional hazards regression model [17] and

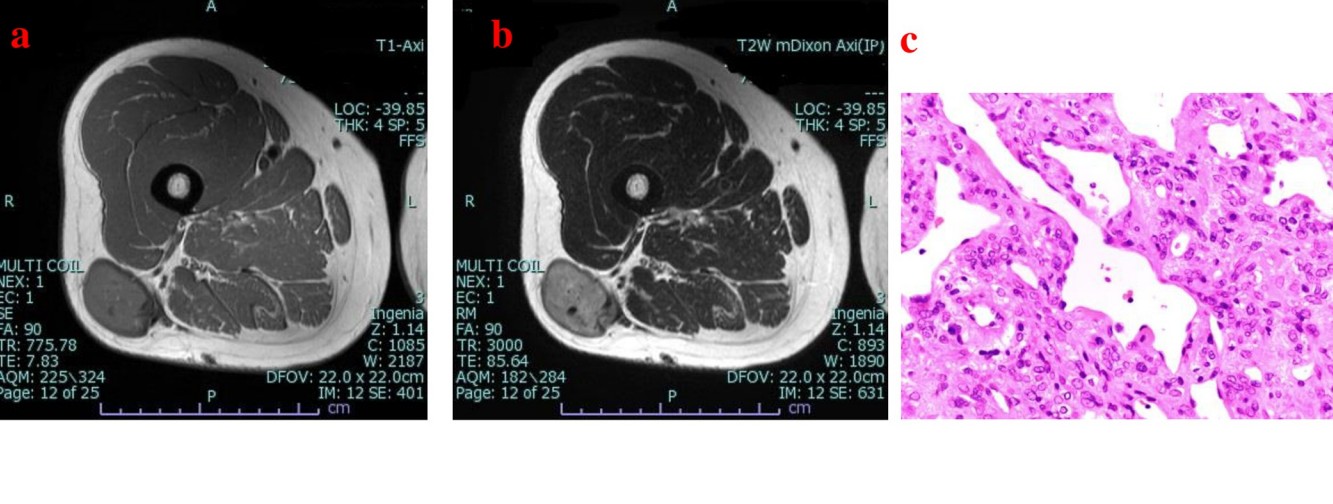

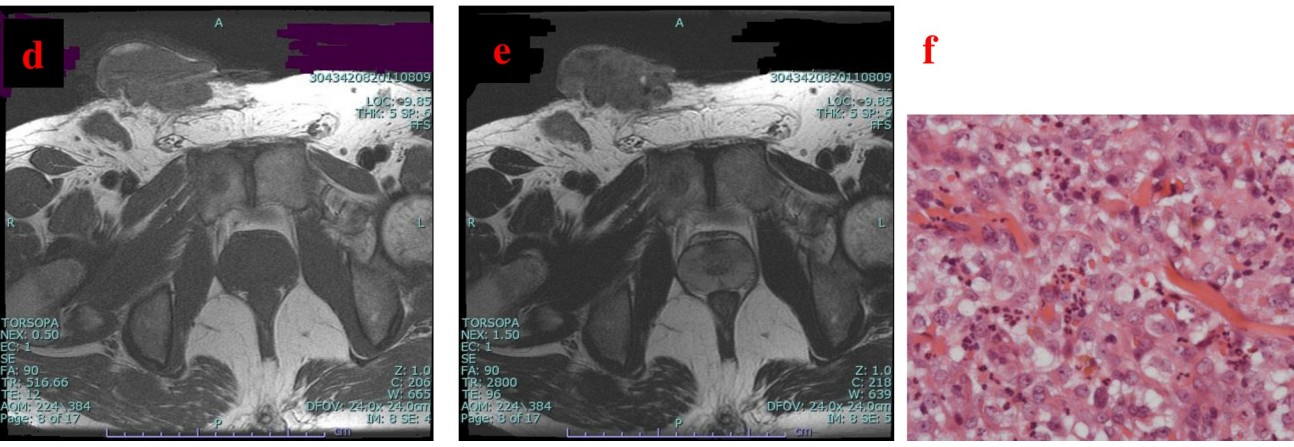

**Fig 2. Representative cases.** (a, b, c) Magnetic resonance imaging (MRI) findings for a superficial lesion in the right thigh in a 44-year-old woman. Axial T1-weighted (a) and T2-weighted (b) images reveal that the lesion did not invade the skin. Pathological examination of the specimen confirmed a solitary fibrous tumor (c) (hematoxylin-eosin staining; magnification ×400). (d, e, f) MRI findings for a superficial lesion in the right inguinal region in a 46-year-old man. Axial T1-weighted (d) and T2-weighted (e) images reveal that the lesion invaded the skin. Pathological examination of the specimen confirmed epithelioid sarcoma (f) (hematoxylin-eosin staining; magnification ×400).

included only two factors (each presenting with $P<0.1$ in the univariate analysis). Fisher's exact probability test was performed to compare the two variables. In all analyses, $P<0.05$ was considered the threshold for statistical significance. All statistical analyses were performed using the Excel statistical software package (Ekuseru-Toukei 2015; Social Survey Research Information Co., Ltd., Tokyo, Japan).

## Results

### Clinical information

The superficial soft tissue sarcomas identified in this study were located in the inferior limbs, trunk, and superior limbs of 40, 27, and 15 patients, respectively; A total of 52 and 30 of the cases presented as high-grade and low-grade tumors, respectively. With regard to tumor size, 44 cases involved a tumor measuring ≥5 cm, of which 31 (70.5%) cases were high-grade sarcomas. The mean tumor size in the current study was 6.0±4.31 cm, and the

**Table 1. Descriptive statistics regarding patients' demographic data.**

| Factors | | Descriptive |
|---|---|---|
| Male | | 48 |
| Female | | 34 |
| Age (years) | | 63.6±17.6 |
| ≥65 | | 49 |
| <65 | | 33 |
| Tumor size (cm) | | |
| Total | | 6.00±4.31 |
| High grade | | 6.02±3.77 |
| Low grade | | 5.97±5.20 |
| Anatomical location | | |
| Extremities | Upper arm | 5 |
| | Forearm | 9 |
| | Hand | 1 |
| | Thigh | 23 |
| | Lower leg | 14 |
| | Foot | 3 |
| Trunk | Chest | 10 |
| | Shoulder | 7 |
| | Back | 1 |
| | Hip | 2 |
| | Abdomen | 4 |
| | Neck | 2 |
| | Head | 1 |
| Tumor grade | High | 52 |
| | Low | 30 |
| | | |
| AJCC stage | I | 29 |
| | II | 27 |
| | III | 14 |
| | IV | 12 |
| Distant metastasis at initial diagnosis | Positive | 12 |
| | Negative | 70 |

Data are presented as means (standard deviations) for continuous variables or counts for categorical variables.

mean sizes of high-grade and low-grade sarcomas were 6.02±3.77 cm and 5.97±5.2 cm, respectively (Table 1).

## Histological diagnoses

Histopathologically, 82 cases of superficial soft tissue sarcomas were recorded (Table 2). The most common types of superficial soft tissue sarcomas identified in the current study were myxofibrosarcoma (13 cases, 16%), pleomorphic liposarcoma (11 cases, 13%), and undifferentiated pleomorphic sarcoma (10 cases, 12%).

We identified four pleomorphic liposarcomas, three undifferentiated pleomorphic sarcomas, one myxofibrosarcoma, one dedifferentiated liposarcoma, one epithelioid sarcoma, one malignant peripheral nerve sheath tumor, and one alveolar soft part sarcoma with skin invasion (Table 2).

**Table 2. Relationship of each histological classification with skin invasion outcomes.**

| | Number | Skin invasion | |
|---|---|---|---|
| | | Positive | Negative |
| Myxofibrosarcoma | 13 | 1 | 12 |
| Pleomorphic liposarcoma | 11 | 4 | 7 |
| Undifferentiated pleomorphic sarcoma | 10 | 3 | 7 |
| Atypical lipomatous tumor | 7 | 0 | 7 |
| Leiomyosarcoma | 6 | 0 | 6 |
| Dedifferentiated liposarcoma | 6 | 1 | 5 |
| Myxoid liposarcoma | 4 | 0 | 4 |
| Epithelioid sarcoma | 4 | 1 | 3 |
| Solitary fibrous tumor | 4 | 0 | 4 |
| Malignant peripheral nerve sheath tumor | 4 | 1 | 3 |
| Synovial sarcoma | 2 | 0 | 2 |
| Low grade fibromyxoid sarcoma | 2 | 0 | 2 |
| Dermatofibrosarcoma | 2 | 0 | 2 |
| Clear cell sarcoma | 1 | 0 | 1 |
| Angiosarcoma | 1 | 0 | 1 |
| Malignant giant cell tumor of soft tissue | 1 | 0 | 1 |
| Ossifying fibro myxoid tumor | 1 | 0 | 1 |
| Atypical fibroxanthoma | 1 | 0 | 1 |
| Cellular angiofibroma | 1 | 0 | 1 |
| Alveolar soft part sarcoma | 1 | 1 | 0 |
| Total number | 82 | 12 | 70 |

Data are presented as counts.

## Treatment

All 82 patients underwent surgical resection; the surgical margins were wide (R0) in 74 cases, marginal (R1) in seven cases, and small (R2) in one case. None of the patients underwent amputation or disarticulation. A total of 14 patients received post-operative radiation therapy, of whom three received heavy particle radiotherapy for local recurrence. Thirteen patients received adjuvant and/or neoadjuvant chemotherapy.

## Oncological outcomes

The mean follow-up time in the current study was 60.1 months (range, 3-208 months). With respect to oncological outcomes at the last follow-up, 31 patients were in a continuous disease-free state, 20 patients showed no evidence of disease, nine patients were alive (though with active disease), 18 patients had died of the disease, and four patients had died of another (comorbid) disease.

## Prognostic factors

Univariate analysis revealed that tumor size, distant metastases at initial diagnosis, and tumor-skin invasion were statistically significant prognostic factors for overall survival rate ($P = 0.015$, $P = 0.006$, and $P<0.001$, respectively) (Table 3). According to the results of the multivariate analysis, the tumor-skin invasion was identified as a statistically significant prognostic factor ($P = 0.008$); in contrast, distant metastasis at initial diagnosis, though showing marginal significance, was not a statistically significant prognostic factor ($P = 0.057$) (Table 3).

**Table 3. Cox proportional hazards analysis for overall survival.**

| Variable (univariate analysis) | Hazard | Referent | HR (95% CI) | P-value |
|---|---|---|---|---|
| Age | ≥65 years | <65 years | 1.49 (0.60–3.67) | 0.39 |
| Sex | Male | Female | 1.22 (0.52–2.87) | 0.65 |
| Location | Trunk | Extremities | 1.99 (0.86–4.63) | 0.11 |
| Size | ≥5 cm | <5 cm | 3.45 (1.27–9.37) | *0.015* |
| Distant metastases upon initial diagnosis | Positive | Negative | 3.50 (1.43–8.56) | *0.006* |
| Grade | High | Low | 2.16 (0.72–6.49) | 0.17 |
| Skin invasion | Positive | Negative | 4.59 (1.89–11.2) | *<0.001* |
| **Variable (multivariate analysis)** | **Hazard** | **Referent** | **HR (95% CI)** | **P-value** |
| Distant metastases upon initial diagnosis | Positive | Negative | 2.50 (0.97–6.39) | 0.0567 |
| Skin invasion | Positive | Negative | 3.55 (1.39–9.03) | *0.008* |

HR, hazard ratio; CI, confidence interval; P-value, P-value from Cox regression analysis

## Survival rates

The 5-year overall survival rate was estimated to be 76.4% using the Kaplan-Meier method (Fig 3); the 5-year local recurrence-free survival and 5-year metastasis survival rates were 60.6% and 71.0%, respectively. There was no statistically significant difference in 5-year overall survival rate between patients aged <65 years and those aged ≥65 years ($P = 0.39$) (Fig 3). Additionally, a statistically significant difference was not identified with regard to tumor grade (low *vs.* high-grade sarcoma; $P = 0.17$) (Fig 3). In contrast, a statistically significant difference was observed for skin invasion ($P<0.001$) (Fig 3), tumor size (≥5 cm *vs.* <5 cm; $P = 0.01$), and distant metastases at initial diagnosis ($P = 0.004$) (Fig 3). The 5-year overall survival rate for patients with skin invasion was lower than that for patients without skin invasion (42.9% *vs.* 82.3%). Moreover, the 5-year overall survival rate for patients with distant metastases at initial diagnosis was lowered as compared with that for patients without metastases (41.7 *vs.* 82.5%).

## Discussion

Previously, we reported that malignant wounds present within soft tissue sarcomas [7]. However, we did not confirm whether malignant wounds in soft tissue sarcomas were related to poor prognoses. A recent study demonstrated that malignant wounds in soft tissue sarcomas were statistically significant poor prognostic factors [9]. While previous reports indicated a better outcome if adequate surgery (i.e., wide resection) was performed in superficial soft tissue sarcomas [2], we hypothesized that skin invasion would likewise be a poor prognostic factor.

The characteristics of superficial soft tissue sarcomas are different from those of deep-seated sarcomas. Soft tissue sarcomas uncommonly manifest superficial to the fascia [18]. In Japan, the incidence rate for soft tissue sarcoma is approximately 2-3/100,000/year [19] and superficial soft tissue sarcomas account for approximately 20% of all soft tissue sarcomas [20]. Superficial soft tissue sarcomas often occur in the extremities, especially in the thigh [21]. According to current medical guidelines, these tumors may be left untreated unless they invade the skin and skin ulcers manifest. However, the longer the duration during which these malignant tumors are left untreated, the less favorable their prognosis would become. To avoid poor oncological outcomes, a rigorous analysis should be conducted with the goal of informing medical guidelines.

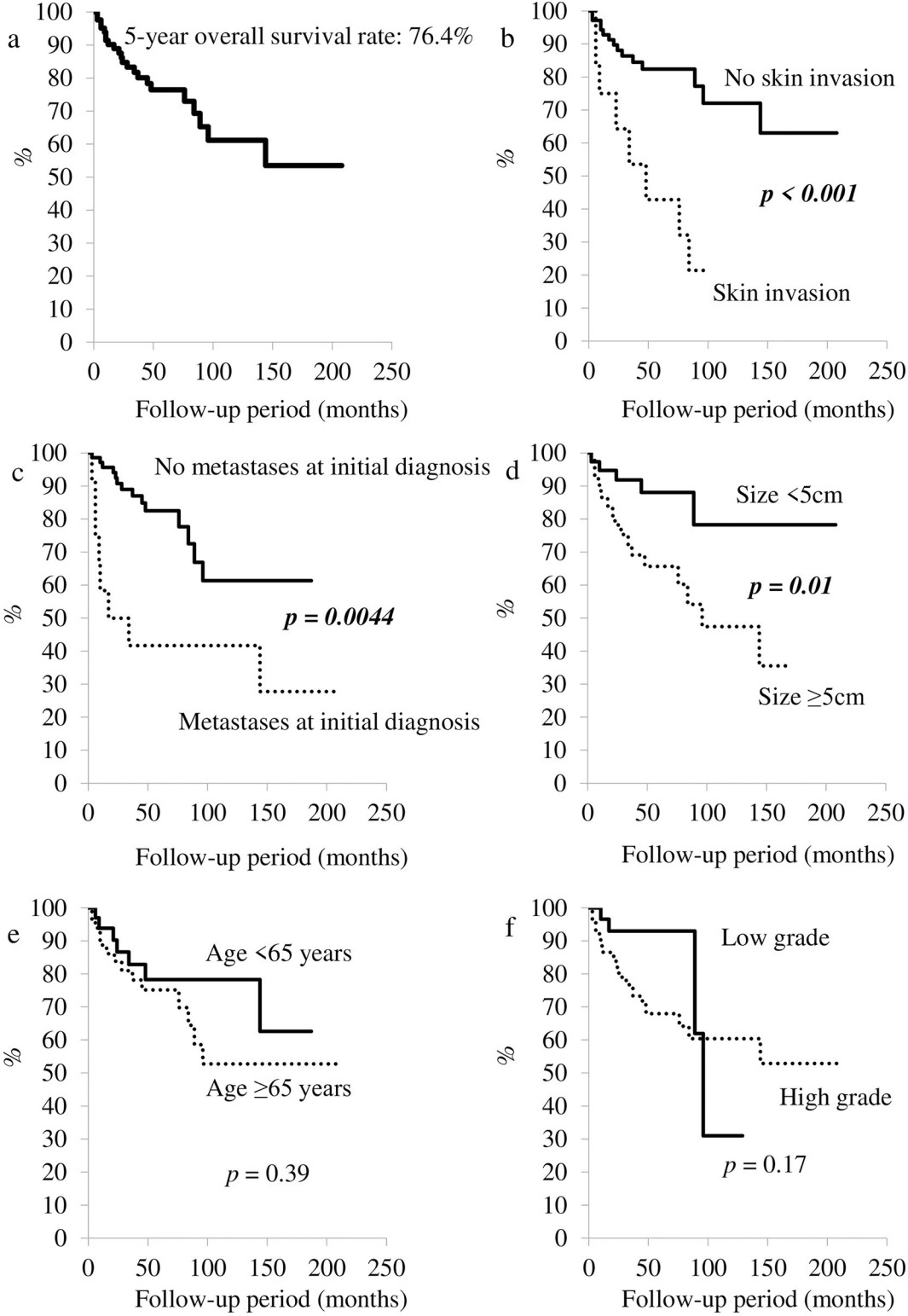

**Fig 3. Survival curves.** a: Overall survival curve. b: Survival curves for patients with sarcomas invading the skin as well as for patients with sarcomas that did not invade the skin. c: Survival curves for patients with metastases and those without metastases upon initial diagnosis. d: Survival curves for patients with sarcomas measuring <5 cm and ≥5 cm. e: Survival curves for patients aged <65 and ≥ 65 years. f: Survival curves for patients with high- and low-grade sarcomas.

**Table 4. Associations with tumor-skin invasion in superficial soft tissue sarcomas.**

| | | Skin invasion | | *P*-value |
|---|---|---|---|---|
| | | **Positive** | **Negative** | |
| Size | | | | |
| | ≥5 cm | 12 | 38 | <*0.001* |
| | <5 cm | 0 | 32 | |
| Distant metastases upon initial diagnosis | | | | |
| | Positive | 5 | 7 | *0.013* |
| | Negative | 7 | 63 | |

P-value, P-value from Fisher's exact probability test

In this study, the most frequent histopathological observations were myxofibrosarcoma (15.9%), pleomorphic liposarcoma (13.4%), and undifferentiated pleomorphic sarcoma (12.2%). Our results mostly coincide with reported general incidence rates according to data from a nationwide registry of soft tissue sarcomas in Japan [19].

In this study, the 5-year overall survival rate for patients with superficial soft tissue sarcomas was 76.4%. Additionally, the 5-year local recurrence-free survival and 5-year metastasis-free survival rates were 60.6% and 71.0%, respectively. Salas et al. reported 5-year overall survival, local recurrence-free survival, and metastasis-free survival rates of 80.9%, 74.7%, and 80.7%, respectively [2]. Our results may have yielded a lower overall survival rate than those of previous reports due to the enrollment of a large number of patients with distant metastases at initial diagnosis in the current study. However, after excluding the cases with distant metastases at initial diagnosis, we reported the 5-year overall survival, local recurrence-free survival, and metastasis-free survival rates of 82.5%, 76.0%, and 83.9%, respectively.

According to the results of our multivariate analysis conducted using Cox proportional hazards regression modeling, the only feature that was highly associated with poor prognoses in the current study was the tumor-skin invasion; while age, tumor size >5 cm, high-grade tumors, and distant metastases at initial diagnosis were not associated with poor prognoses. Age has been shown to be a poor prognostic factor within soft tissue sarcomas [22]; this is likely because of the lowered physical function and general tolerance to treatment among aged patients. However, with the advent of multimodal medical technologies, aged patients would be able to undergo surgery with a lower risk [23, 24]. The present study demonstrated a lack of statistically significant survival differences between patients aged <65 years and those aged ≥65 years.

It is well recognized that soft tissue tumors measuring more than 5 cm are a statistically significant prognostic factor when differentiating between malignant and benign lesions in the superficial subset [3], and a previous study indicated that almost all superficial soft tissue sarcomas were ≥5 cm in size [21]. We also previously showed that tumors measuring ≥5 cm as well as the tumor-fascia relation upon MRI were statistically significant indicators of malignancy [25]. The present study revealed that superficial soft tissue sarcomas measuring ≥5 cm were not associated with poor prognoses. However, all 12 cases with tumor-skin invasion enrolled in the current study had tumor sizes of 5 cm or more. We found a statistically significant association between the tumor-skin invasion and tumor size (P<0.001) (Table 4). Based on this result, we conclude that sarcomas with the tumor-skin invasion are likely to measure ≥5 cm.

In this study, patients with high-grade soft tissue sarcomas were diagnosed as stage II or III according to the American Joint Committee on Cancer guidelines (8<sup>th</sup> edition) with respect to

soft tissue tumors and bone cancer [13]. Several studies report that high-grade sarcomas may aggressively infiltrate into adjacent tissues, resulting in poor prognoses [26, 27]. However, we did not find statistically significant associations between tumor grade and poor prognoses.

In addition, sarcomas presenting with distant metastases at initial diagnosis were defined in this study as stage IV in accordance with the 8[th] edition of the American Joint Committee on Cancer guidelines for soft tissue tumors and bone cancer [13]. Based on this diagnosis, distant metastasis was previously reported as the poorest prognostic factor among all evaluated factors [26, 27]; however, the present report showed that the presence of distant metastases was not related to poor prognoses. This may be because our study was limited in statistical power due to its small sample size. Therefore, we recommend that a multivariate analysis should be reperformed in the future. However, five of the 12 cases with tumor-skin invasion enrolled in the current study were also initially diagnosed with metastatic sarcoma and we found a statistically significant association between tumor-skin invasion and distant metastases upon initial diagnosis ($P$ = 0.013) (Table 4). Based on these findings, we infer that sarcomas with tumor-skin invasion are more likely to present with distant metastases.

Benign tumors often enlarge to compress the adjacent tissues, including benign tumors in the skin; however, benign tumors are unlikely to invade the adjacent tissues. In contrast, malignant tumors tend to extend along the skin and occasionally infiltrate the skin. According to the conceptualization of surgical margins for bone and soft tissue sarcoma, skin can be thought of as a barrier [14]. Therefore, skin invasion in superficial soft tissue tumors can be indicative of locally aggressive malignant tumors, though few reports have found associations between skin invasion and poor prognoses. The present study focused on superficial soft tissue sarcomas, with multivariate analysis conducted to verify statistically significant poor prognostic factors. Thus, the tumor-skin invasion was identified as a prognostic factor in the current study, while distant metastasis upon initial diagnosis was not.

In addition to the substantial strengths of our investigation, we acknowledge several limitations, including the small number of cases enrolled in our study as well as our retrospective study design. Our results should be verified in a study involving a larger number of cases and multiple centers. Second, this study included patients with various stages of sarcoma, histological types, and tumor grades and locations. In particular, the percentage of myxofibrosarcoma identified during data collection was high. In general, myxofibrosarcoma represents approximately 20% of all soft tissue sarcomas, especially among elderly patients [28]. Third, superficial soft tissue sarcomas were defined based on their location (i.e., skin, subcutaneous). However, there are no standard criteria defining superficial tumor locations within current medical guidelines. Hence, we evaluated the locations of the lesions by examining each MRI image separately; lesions were defined as superficial soft tissue sarcomas if the percentage of the area occupied by the tumor in the assessed region (cutaneous, subcutaneous) was more than 50% (Fig 1A). Thus, the resulting differential or non-differential misclassification is likely to have been a major limitation of the present study. Fourth, we did not examine the immunological and molecular mechanisms of the tumor-skin invasion. Therefore, it will be indispensable for further research to elucidate the relationship between invasion and poor prognosis in the future.

## Conclusion

Only tumor-skin invasion was closely associated with poor prognoses within superficial soft tissue masses in the current analysis. If primary superficial soft tissue sarcomas invade the skin, large tumors measuring ≥5 cm as well as the presence of distant metastases should be suspected. In this study, no statistically significant correlations were observed between distant

metastases upon initial diagnosis and poor prognoses. However, based on the results of the current study and the general literature to date, we strongly recommend that superficial soft tissue sarcomas should be treated before skin invasion and skin ulcers manifest, regardless of the patient's age, tumor size, tumor grade, or the presence of distant metastases upon initial diagnosis.

Not only further prospective studies enrolling a larger number of patients and involving multiple centers but also strategic basic research, such as immunological and molecular evaluation is warranted to substantiate the relationship between tumor-skin invasion, distant metastasis, and poor prognosis.

## Supporting information

**S1 Table. Anonymized minimal data set.**
(PDF)

## Author Contributions

**Conceptualization:** Tadashi Iwai.

**Data curation:** Tadashi Iwai, Naoto Oebisu, Naoki Takada, Yoshitaka Ban.

**Formal analysis:** Tadashi Iwai.

**Investigation:** Tadashi Iwai.

**Project administration:** Manabu Hoshi.

**Supervision:** Manabu Hoshi, Hiroaki Nakamura.

**Validation:** Tadashi Iwai.

**Visualization:** Tadashi Iwai.

**Writing – original draft:** Tadashi Iwai.

**Writing – review & editing:** Manabu Hoshi.

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
