## [Decision Letter · Decision Letter 0]

8 Aug 2022

PONE-D-22-19625Tumor-skin Invasion is a Reliable Risk Factor for Poor Prognosis in Superficial Soft Tissue SarcomasPLOS ONE

Dear Dr. Iwai,

Thank you for submitting your manuscript to PLOS ONE. After careful consideration, we feel that it has merit but does not fully meet PLOS ONE’s publication criteria as it currently stands. Therefore, we invite you to submit a revised version of the manuscript that addresses the points raised during the review process.

If you will need more time than this to complete your revisions, please reply to this message or contact the journal office at plosone@plos.org. Please include the following items when submitting your revised manuscript:A rebuttal letter that responds to each point raised by the academic editor and reviewer(s). You should upload this letter as a separate file labeled 'Response to Reviewers'.A marked-up copy of your manuscript that highlights changes made to the original version. You should upload this as a separate file labeled 'Revised Manuscript with Track Changes'.An unmarked version of your revised paper without tracked changes. You should upload this as a separate file labeled 'Manuscript'.If applicable, we recommend that you deposit your laboratory protocols in protocols.io to enhance the reproducibility of your results. Protocols.io assigns your protocol its own identifier (DOI) so that it can be cited independently in the future. For instructions see: https://journals.plos.org/plosone/s/submission-guidelines#loc-laboratory-protocols. Additionally, PLOS ONE offers an option for publishing peer-reviewed Lab Protocol articles, which describe protocols hosted on protocols.io. Read more information on sharing protocols at https://plos.org/protocols?utm_medium=editorial-email&utm_source=authorletters&utm_campaign=protocols.

We look forward to receiving your revised manuscript.

Kind regards,

Filomena de Nigris, M.D., Ph.D.

Academic Editor

PLOS ONE

Journal Requirements:

2. In the ethics statement in the Methods and online submission information, please ensure that you have specified (1) whether consent was informed and (2) what type you obtained (for instance, written or verbal, and if verbal, how it was documented and witnessed). If your study included minors, state whether you obtained consent from parents or guardians. If the need for consent was waived by the ethics committee, please include this information.

4. Please upload a copy of Figures 2 and 3 to which you refer in your text on pages 7 and 15. If the figure is no longer to be included as part of the submission please remove all reference to it within the text.

Reviewers' comments:

Reviewer's Responses to Questions

**Comments to the Author**

1. Is the manuscript technically sound, and do the data support the conclusions?

The manuscript must describe a technically sound piece of scientific research with data that supports the conclusions. Experiments  have been conducted rigorously, with appropriate controls, replication, and sample sizes. The conclusions be drawn appropriately based on the data presented. 

Reviewer #1: Yes

Reviewer #2: Partly

2. Has the statistical analysis been performed appropriately and rigorously? 

Reviewer #1: Yes

Reviewer #2: Yes

3. Have the authors made all data underlying the findings in their manuscript fully available?

Reviewer #1: Yes

Reviewer #2: Yes

4. Is the manuscript presented in an intelligible fashion and written in standard English?

Reviewer #1: Yes

Reviewer #2: No

5. Review Comments to the Author

Reviewer #1: I do not have many suggestions regarding the paper itself, it is quite well written, with good literature review.

The quality of figures should be improved.

I would suggest to cite this recent review: Oettel, D.J., Bernard, S.A. Review of primary superficial soft tissue mesenchymal tumors of malignant or intermediate biological potential. Skeletal Radiol (2022). https://doi.org/10.1007/s00256-022-04127-0

Reviewer #2: In the manuscript entitled “Tumor-skin Invasion is a Reliable Risk Factor for Poor Prognosis in Superficial Soft Tissue Sarcomas” the authors describe that the tumor-skin invasion is closely associated with poor prognoses.

The manuscript is well written and the data are presented with great accuracy. However, I have some minor suggestions to share with the authors:

• I suggest the authors provide the statistical analysis used in the legend of tables.

• The "Introduction" section is poor in information. the authors have to expand the text with the immunological aspect of sarcoma. For this aspect, I suggest citing the article "Sarcoma Common MHC-I Haplotype Restricts Tumor-Specific CD8+ T Cell Response. 2022, " ext-link-type="uri" xlink:type="simple">doi.org/10.3390/cancers14143414".

• In addition, the authors could discuss the mechanisms of invasion.

• Finally, the future perspectives of the authors must be more elaborated.

• English revision is required.

Reviewer #1: No

Reviewer #2: No

---

## [Author Response · Author response to Decision Letter 0]

18 Aug 2022

Journal Requirements:

Response: We thank the editor for these suggestions. We have gone through these templates and ensured that the style requirements of the PLOS ONE are met by making appropriate stylistic corrections in the revised manuscript.

2. In the ethics statement in the Methods and online submission information, please ensure that you have specified (1) whether consent was informed and (2) what type you obtained (for instance, written or verbal, and if verbal, how it was documented and witnessed). If your study included minors, state whether you obtained consent from parents or guardians. If the need for consent was waived by the ethics committee, please include this information.

Response: We thank the editor for these suggestions. Our research was a retrospective study, which was approved by the Institutional Review Board of Osaka Metropolitan University Graduate School of Medicine and was performed in accordance with the ethical standards laid down in the Declaration of Helsinki (no. 4394). This information is a part of the methods section of the manuscript (Lines 80-84).

Response: We thank the editor for these insights. We have added and uploaded the minimal anonymized data set as supporting information (Page 26, lines 455-456).

4. Please upload a copy of Figures 2 and 3 to which you refer in your text on pages 7 and 15. If the figure is no longer to be included as part of the submission please remove all reference to it within the text.

Response: We thank the editor for pointing this out. We have uploaded the copies of Figures 2 and 3 in the revised submission.

Response: We thank the editor for their advice. We have rechecked our reference list and corrected that in the revised manuscript.

Review Comments to the Author

Reviewer #1: I do not have many suggestions regarding the paper itself, it is quite well written, with good literature review.

The quality of figures should be improved.

Response: We thank the reviewer for evaluating our manuscript and for these valuable suggestions. We have improved the quality of figures and uploaded them in the revised submission.

I would suggest to cite this recent review: Oettel, D.J., Bernard, S.A. Review of primary superficial soft tissue mesenchymal tumors of malignant or intermediate biological potential. Skeletal Radiol (2022). https://doi.org/10.1007/s00256-022-04127-0

Response: We thank the reviewer for their advice. We believe that citing this review has indeed improved our manuscript. We have cited this article on Page 4, lines 46-48.

Page 4, lines 46-48: However, when the tumors occur superficially, that might distort the clinical decision-making of the oncologists and thus, they might not suspect sarcomas, resulting in a misdiagnosis [5].

Reviewer #2: In the manuscript entitled “Tumor-skin Invasion is a Reliable Risk Factor for Poor Prognosis in Superficial Soft Tissue Sarcomas” the authors describe that the tumor-skin invasion is closely associated with poor prognoses.

The manuscript is well written and the data are presented with great accuracy. However, I have some minor suggestions to share with the authors:

• I suggest the authors provide the statistical analysis used in the legend of tables.

Response: We thank the reviewer for a careful review of our manuscript and for their helpful suggestions. We have added the statistical analysis used in the legend of tables. Please refer to Table 3a (Cox regression analysis) and Table 3b (Fisher’s exact probability test).

• The "Introduction" section is poor in information. the authors have to expand the text with the immunological aspect of sarcoma. For this aspect, I suggest citing the article "Sarcoma Common MHC-I Haplotype Restricts Tumor-Specific CD8+ T Cell Response. 2022, doi.org/10.3390/cancers14143414".

Response: We thank the reviewer for highlighting this gap and for suggesting us the remedy. We believe that including this citation has certainly strengthened the manuscript. We have cited this article on Page 4, lines 58-60.

Page 4, lines 58-60: even though cutting-edge research indicates that an assessment of the major histocompatibility complex is crucial for the clinical outcome of sarcoma immunotherapy [10].

• In addition, the authors could discuss the mechanisms of invasion.

Response: We thank the reviewer for this kind advice. We have added the details and believe that this has helped us to improve the manuscript (Page 21, lines 335-338).

Page 21, lines 335-338: Fourth, we did not examine the immunological and molecular mechanisms of tumor-skin invasion. Therefore, it will be indispensable for further research to elucidate the relationship between invasion and poor prognosis in the future.

• Finally, the future perspectives of the authors must be more elaborated.

Response: We thank the reviewer for these insights. We have included future perspectives based on the reviewer’s suggestions (Page 21, lines 351-354).

Page 21, lines 351-354: Not only further prospective studies enrolling a larger number of patients and involving multiple centers but also strategic basic research, such as immunological and molecular evaluation is warranted to substantiate the relationship between tumor-skin invasion, distant metastasis, and poor prognosis.

• English revision is required.

Response: We thank the reviewer for this helpful advice. We have substantially improved the quality of English used in our manuscript.

---

## [Editor Report · Decision Letter 1]

23 Aug 2022

Tumor-skin invasion is a reliable risk factor for poor prognosis in superficial soft tissue sarcomas

PONE-D-22-19625R1

Dear Dr.Tadashi Iwai

We’re pleased to inform you that your manuscript has been judged scientifically suitable for publication and will be formally accepted for publication once it meets all outstanding technical requirements.

Kind regards,

Filomena de Nigris, Ph.D.

Academic Editor

PLOS ONE
---

## [Editor Report · Acceptance letter]

25 Aug 2022

PONE-D-22-19625R1 

Tumor-skin invasion is a reliable risk factor for poor prognosis in superficial soft tissue sarcomas 

Dear Dr. Iwai:

I'm pleased to inform you that your manuscript has been deemed suitable for publication in PLOS ONE. Congratulations! Your manuscript is now with our production department. 

Kind regards, 

on behalf of

Prof. Filomena de Nigris 

Academic Editor

PLOS ONE